# Antiretroviral treatment failure and associated factors among people living with HIV on therapy in Homa Bay, Kenya: A retrospective study

**Rose Masaba**[1]*, **Godfrey Woelk**[2], **Stephen Siamba**[1], **James Ndimbii**[1], **Millicent Ouma**[1], **Jacob Khaoya**[1], **Abraham Kipchirchir**[1], **Boniface Ochanda**[3], **Gordon Okomo**[4]

**1** Elizabeth Glaser Pediatric AIDS Foundation, Nairobi, Kenya, **2** Elizabeth Glaser Pediatric AIDS Foundation, Washington, D.C., United States of America, **3** Division of Global HIV & TB, Center for Global Health, US Centers for Disease Control and Prevention Kenya, Kisumu, Kenya, **4** Ministry of Health, Homa Bay County, Kenya

* rmasaba@pedaids.org

**Data Availability Statement:** The dataset generated and/or analyzed during the current study

## Abstract

Despite large numbers of patients accessing antiretroviral treatment (ART) in Kenya, few studies have explored factors associated with virologic failure in Western Kenya, specifically. We undertook a study in Homa Bay County, Kenya to assess the extent of virologic treatment failure and factors associated with it. This was an observational retrospective study conducted from September 2020 to January 2021. Data were abstracted from the records of patients who had been on ART for at least six months at the time of data collection after systematic sampling stratified by age group at ART initiation (0–14 and 15+ years), using probability proportion to the numbers of patients attending the facility. Confirmed viral treatment failure was defined as viral load ≥1000 copies/ml based on two consecutive viral load measurements after at least three months of enhanced adherence counseling. Data were analyzed using descriptive statistics and Cox regression modeling. Of the 2,007 patients sampled, 160 (8.0%) had confirmed virologic treatment failure. Significantly higher virologic treatment failure rates were identified among male patients 78/830 (9.4%) and children 115/782 (14.7%). Factors associated with virologic treatment failure (VTF), were age 0–14 years, adjusted hazard ratio (AHR) 4.42, (95% Confidence Interval [CI], 3.12, 6.32), experience of treatment side effects AHD: 2.43, (95% CI, 1.76, 3.37), attending level 2/3 health facility, AHR: 1.87, (95% CI, 1.29, 2,72), and history of opportunistic infections (OIs), AHR: 1.81, (95% CI, 1.76, 3.37). Children, attendees of level 2/3 health facilities, patients with a history of OIs, and those experiencing treatment side-effects are at risk of VTF. Increased focus on children and adolescents on screening for drug resistance, administration of and adherence to medication, and on effective information and education on side-effects is critical. Additionally, there is need for increased training and support for health care workers at primary level care facilities.

has been uploaded as supplementary material with this manuscript.

**Funding:** This study was supported by the U.S. President's Emergency Plan for AIDS Relief (PEPFAR) through the U.S. Centers for Disease Control and Prevention (CDC) under the terms of Cooperative Agreement No. NU2GGH001948-01-00. The funder contributed to study design and reviewed the manuscript and did not interact with human subjects or have access to identifiable data or specimens for research purposes. The funder cleared the manuscript for publication. The findings and conclusions in this manuscript are those of the author(s) and do not necessarily represent the official position of the Centers for Disease Control and Prevention and other funding agencies.

**Competing interests:** The authors have declared that no competing interests exist.

## Introduction

The global scale-up of life-saving antiretroviral treatment (ART) was a critical turning point in the clinical management of HIV and reduction in AIDS-related deaths. Worldwide, there were an estimated 38.4 million people living with HIV (PLHIV) in 2021, of whom roughly 28.7 million were accessing ART in 2021 [1]. Although the proportion of PLHIV accessing ART has greatly increased in recent years, the UNAIDS 95-95-95 testing and treatment targets are yet to be achieved. According to UNAIDS 2021 data, 85% of PLHIV knew their status, 75% were accessing treatment, and 68% were virally suppressed [1].

In Kenya, the national adult HIV prevalence rate is estimated at 4.0%, or 1.4 million PLHIV aged 15 years and older, in addition to 83,000 children aged 0–14 years living with HIV [2]. There were 1.1 million PLHIV on ART in 2021, with 79% and 59% treatment coverage among adults and children, respectively [2]. However, the effectiveness of national ART efforts can be compromised if patients are not properly monitored and managed.

As more patients are initiated on ART, the number of people on treatment who are not adhering to ART or appropriately changing regimens after virologic treatment failure (VTF) subsequently increases, as does the number of PLHIV on second- and third-line regimens [3, 4]. VTF is the sub-optimal response or the lack of sustained response to ART and can be measured in three ways: (1) clinically, by disease progression or a new or recurrent clinical event indicating an advanced or severe immune deficiency (World Health Organization (WHO) clinical stage III and IV clinical conditions); (2) immunologically, by a fall of CD4 count to the baseline or persistent CD4 levels below 200 cells/ml; and (3) virologically, by plasma viral load (VL) $\geq$1000 copies/ml based on two consecutive VL measurements in three months, with enhanced adherence counselling (EAC) following the first VL test [5]. EAC is a targeted counseling strategy designed to help clients identify individual barriers to adherence and develop strategies to improve viral suppression. The EAC package is offered in at least three sessions. The first counseling session focuses on treatment literacy for adolescents and caregivers for younger children. During this session, the adherence counselor supports the client to identify individual barriers to optimal ART adherence. The counselor then supports the client to develop relevant strategies to overcome the barriers through development of an adherence plan. ART adherence plans are reviewed by the counselor and adjusted jointly with the client in the second and third sessions, with any emerging issues discussed. To prepare for EAC sessions, providers are trained on psychosocial support and communication skills for children living with HIV and their caregivers.

The WHO recommends using VL testing as the preferred method to diagnose and confirm VTF [6]. Similarly, Kenya's Ministry of Health guidelines use VL as the test of choice to monitor ART response and identify VTF [7]. HIV VTF increases the risk of HIV transmission, therefore access to routine VL monitoring is critical to treatment success.

Previous studies have identified poor ART adherence, prolonged ART use, suboptimal ART regimens, tuberculosis (TB) co-infection, treatment interruption, opportunistic infections (OIs), rural residency, advanced WHO stage, and low baseline CD4 count as factors associated with an increased risk of HIV VTF [8–14]. Adherence to ART has been shown to be one of the most important predictors of virologic success and preventing disease progression [15].

The identification and management of ART failure is a critical component of HIV management. Prolonged duration on a failing regimen increases the risk of drug resistance and potentially the risk of morbidity and mortality [16–18]. Accurately identifying, understanding, and addressing these factors is crucial in achieving viral suppression, reducing the transmission of HIV, and progressing towards epidemic control. Despite the large number of patients accessing ART in Kenya, few studies have explored VTF and its associated factors in this region.

The aim of this study was to determine the VTF rates among patients in Homa Bay County by assessing the outcomes of treatment regimens, regimen switches, and factors associated with VTF.

## Materials and methods

### Study design

This was an observational retrospective cohort study. The study included secondary analysis of routinely-collected HIV treatment individual-level data of clients active on antiretroviral therapy as at December 2019. Outcomes of interest included viral suppression, retention in care, loss to follow up and treatment failure rates.

### Study setting and population

This study was conducted in Homa Bay, a county in western Kenya with a high HIV prevalence of 19.6% [19], compared to the national prevalence of 4.2% [2]. Nationally, the county contributes 9.9% (128,199) of adults and 7.7% (10,722) of children living with HIV. Treatment coverage in Homa Bay County is 91% among adults aged 15 years and above and 75% among children, 0–14 years old. Of the 52,800 national new HIV infections across all ages, 4,558 occurred in Homa Bay County in 2020 [20]. This study was conducted in nine sites purposely selected to represent the Kenya Essential Package for Health levels 2–5 and included dispensaries, health centers (levels 2 and 3, five facilities), and sub-county (level 4, two facilities) and county referral hospitals (level 5, one facility) [21]. All facilities had access to routine VL monitoring and had the highest numbers of patients with unsuppressed VLs within each facility level. At the time of this study, the Elizabeth Glaser Pediatric AIDS Foundation (EGPAF) was supporting the implementation of HIV services in these facilities in collaboration with the Homa Bay County government.

### Sampling and sample size

The study enrolled PLHIV who were active on treatment, had been on treatment for at least six months at the time of data collection and had a viral load done. Based on WHO 2016 guidelines, this time was to enable patients to be stabilized on treatment and have at least one viral load done; viral loads would usually be routinely monitored after 6 months of initiation of ART (5). The sample was stratified by age group, (0–14 years and 15+ years), using probability proportion to the numbers of patients attending the facility. We aimed for a convenient sample size of 2,007, which was considered feasible and achievable given the time and resources available.

### Data collection

Data were collected from September 2020 to February 2021 for patients who were active in care as at February 2021. We used electronic medical records to derive the sample frame from which the sample was drawn. In each facility, systematic sampling was used to select patient files for abstraction, where every nth record (for example, every 10th file), was selected until the sample size allotted for that facility was obtained. The starting point was obtained by randomly sampling a number between 1 and n, i.e., 1 and 10. Data were abstracted by trained research assistants from both electronic and paper-based individual patient files, ART registers, laboratory registers, and clinical cards at the participating facilities. The data were then recorded into an ODK-X data collection tool in a password-protected database.

We abstracted demographic and clinical data from patient files at the sampled health facilities. The collected demographic data included: age, sex, education, and marital status. The clinical data included: WHO stage, CD4 and body mass index (BMI) at ART initiation, ART regimen at initiation, current ART regimen for those with VTF, time to ART initiation following HIV-positive diagnosis, duration on ART, OIs including TB, ART side effects and toxicities, VL, timing of services including uptake and completion of EAC sessions, regimen switches, and treatment outcomes.

For the purposes of this study, VTF was defined as VL of $\geq$1000 copies/ml in a patient who had been on ART for at least six months; suspected VTF (sVTF) was where at least one VL result $\geq$1000 copies/ml was recorded for a patient. Confirmed VTF was a VL result $\geq$1000 copies/ml based on two consecutive VL measurements after at least three months of EAC sessions, time within which barriers to adherence would have been addressed, in accordance with the Kenya guidelines.

## Statistical analysis

We summarized categorical variables using frequencies, percentages/proportions, and continuous variables using medians and interquartile ranges (IQR) and mean and standard deviations, depending on their distributions. The Cox Proportional Hazards regression method was used to estimate the probabilities of VTF. Kaplan-Meier analysis was performed to estimate cumulative incidence of VTF following ART initiation and Log-Rank test was used to assess difference in time to VTF between age groups. Univariate and multivariate statistical significance was assessed at 0.25 and 0.05 levels, respectively, and multivariable hazard regression modeling was used to identify factors associated with VTF using hazard ratios (HRs). We included variables at the 0.25 level of significance in the multivariate analysis. Analysis was undertaken in STATA version 16.0.

## Ethical approval

The study was approved by Kenyatta National Hospital-University of Nairobi Ethics Research Committee and Advarra Institution Review Board in the US. The protocol was also reviewed in accordance with the US Centers for Disease Control and Prevention (CDC) human research protection procedures and was determined to be research, but CDC investigators did not interact with human subjects or have access to identifiable data or specimens for research purposes.

We received a waiver of informed consent for the use of retrospective data. All data were kept confidential and only the study team had access to patient data.

## Results

### Socio-demographic and clinical characteristics of patients

This analysis included data for 2,007 adults and children living with HIV receiving ART. Table 1 shows the demographic and clinical characteristics of those enrolled in the study by age group. Of all study participants, 782 (39.0%) were children, 0–14 years old. A total of 1,165 (58.4%) were female. Overall, the participants had a median age at ART initiation of 25 years (IQR: 6, 36); 4 years (IQR: 1, 8) for children and 33 years (IQR: 27, 42) for adults. About half of all participants (48.2%) were married. Of the 782 children, 10 were living with partner and two were divorced/separated/widowed. The majority (87.5%) of those between 6–18 years of age were in school at the time of ART initiation.

**Table 1. Demographic and clinical characteristics of study participants by age group.**

| Factor | Level | Ages 0–14 N = 782 | Ages 15+ N = 1225 | Total |
|---|---|---|---|---|
| | | n (%) | n (%) | N = 2007 |
| | | | | n (%) |
| Age, years | Median [IQR] | 4 [1, 8] | 33 [27, 42] | 25 [6, 36] |
| Sex[a] | Male | 394 (50.7) | 436 (35.8) | 830 (41.6) |
| | Female | 383 (49.3) | 782 (64.2) | 1165 (58.4) |
| Marital status at ART initiation[b] | Married/living with partner | 10 (1.3) | 955 (78.2) | 965 (48.2) |
| | Not married | 772 (98.7) | 267 (21.8) | 1039 (51.8) |
| WHO stage at ART initiation[c] | WHO I/II | 570 (73.3) | 843 (68.9) | 1413 (70.6) |
| | WHO III/IV | 208 (26.7) | 381 (31.1) | 589 (29.4) |
| Time to ART initiation, days | Median [IQR] | 42 [0, 234] | 59 [0, 407] | 51 [0, 332] |
| | 0–30 | 360 (46.0) | 487 (39.8) | 847 (42.2) |
| | >30 | 422 (54.0) | 738 (60.2) | 1160 (57.8) |
| ART regimen at initiation[c] | NRTI/NNRTI-based | 644 (82.8) | 1144 (93.5) | 1788 (89.3) |
| | PI-based | 130 (16.7) | 28 (2.3) | 158 (7.9) |
| | DTG-based | 4 (0.5) | 52 (4.2) | 56 (2.8) |
| Facility level | Level 2/3 | 172 (22.0) | 194 (15.8) | 366 (18.2) |
| | Level 4/5 | 610 (78.0) | 1031 (84.2) | 1641 (81.8) |
| CD4 at ART initiation, cells/ml of blood[d] | n, median [IQR] | 419, 576 [283, 1009] | 861, 298 [167, 502] | 1280, 360 [200, 618] |
| | <200 | 60 (14.3) | 259 (30.1) | 319 (24.9) |
| | ≥200 | 359 (85.7) | 602 (69.9) | 961 (75.1) |
| Experienced at least one ART side effect | Yes | 153 (19.6) | 268 (21.9) | 421 (21.0) |
| Experienced at least one OI | Yes | 168 (21.5) | 251 (20.5) | 419 (20.9) |
| BMI (kg/m$^2$) at ART initiation[e] | n, median [IQR] | 498, 16 [14, 18] | 1030, 21 [19, 23] | 1528, 19.8 [16.6, 22.5] |
| Schooling[f] | Attending school | 240 (87.9) | 4 (66.7) | 244 (87.5) |
| TB | Screened | 778 (99.5) | 1221 (99.7) | 1999 (99.6) |
| | IPT | 748 (96.1) | 1199 (98.2) | 1947 (97.4) |
| | Diagnosed with TB | 4 (0.5) | 8 (0.7) | 12 (0.6) |
| Confirmed VTF | Confirmed VTF | 115 (14.7) | 45 (3.7) | 160 (8.0) |

Socio-demographic and clinical characteristics of clients disaggregated by age. These client characteristics were described with the age classified as 0–14 and 15+ years

DTG, dolutegravir; IPT, isoniazid preventive therapy; NRTI, nucleoside reverse transcriptase inhibitor; NNRTI, non-nucleoside reverse transcriptase inhibitor; PI, protease inhibitor.

[a] Data were missing for five (aged 0–14) and seven (aged 15+) clients

[b] Data were missing for three (aged 15+) clients

[c] Data were missing for four (aged 0–14) and one (aged 15+) clients

[d] Data were missing for 363 (aged 0–14) and 364 (aged 15+) clients

[e] Excludes children below six years of age

[f] School attendance was only analyzed for clients between 6–18 years of age

At the time of ART initiation, a quarter of participants (319 [24.9%]) aged six years or older, had advanced HIV disease (CD4 count <200 cells/mL). Children had a median CD4 of 576 cells/mL (IQR: 283, 1009), while adults had median of 298 cells/mL (IQR: 167, 502). The majority of the participants (89.3%) were initiated on NNRTI-based regimens. The median time from diagnosis to ART initiation was 51 days (IQR: 0, 332). Almost all participants, 99.6%, had been screened for TB and 1,947 (97.4%) of those who screened negative had received IPT.

## Treatment failure

Using virologic criteria, at least 280/2007 (13.9%) of the participants experienced one episode of sVTF (Fig 1). Of these participants, EAC uptake was 248/280 (88.6%). All the 248 participants had a repeat VL done. A total of 254 participants had a repeat VL, six of whom had no record of receiving EAC sessions. Overall, 160/2007 (8.0% [95% CI: 6.8%, 9.3%]) were confirmed with VTF following EAC and a repeat VL after three months. Of the 160 with confirmed VTF, 121(84.2%), completed three sessions of EAC (Male, 57/78 (73%), Female, 64/79 (81%)). Of the six who had no documentation of EAC, but had a repeat VL, three were confirmed with VTF. When disaggregated by age at ART initiation, the VTF rate was 14.7% (95% CI: 12.2%, 17.2%) and 3.7% (95% CI: 2.6%, 4.7%) among those aged 0–14 and 15+ years at time of ART initiation, respectively (p<0.001). At the time of data collection, 63/160 (39.4%) of participants with confirmed VTF had not been switched to a different ART regimen (Fig 1). The majority, 84/160 (52.5%), of VTFs occurred between 24 and 60 months after ART initiation (Fig 2). The overall median time to VTF was 65 (IQR: 41, 97) months after ART initiation; 62 (IQR: 40, 89) and 76 (IQR: 50, 112) among patients aged 0–14 years and 15+ years, respectively (p = 0.0431) (Fig 3).

The median age at ART initiation for those with confirmed VTF was 13.0 years (IQR: 8, 29). About three-quarters of the participants with confirmed VTF, 115 (71.9%), were 0–14 years old at ART initiation. At the time of diagnosis of VTF, 89.4% of the participants were on NRTI/NNRTI-based ART regimens and about one third had at least one reported OI. The majority of participants with VTF had been screened for TB and seven had been diagnosed with and treated for TB. Additionally, 40% of participants with VTF had experienced side effects and toxicities to ART (Table 2).

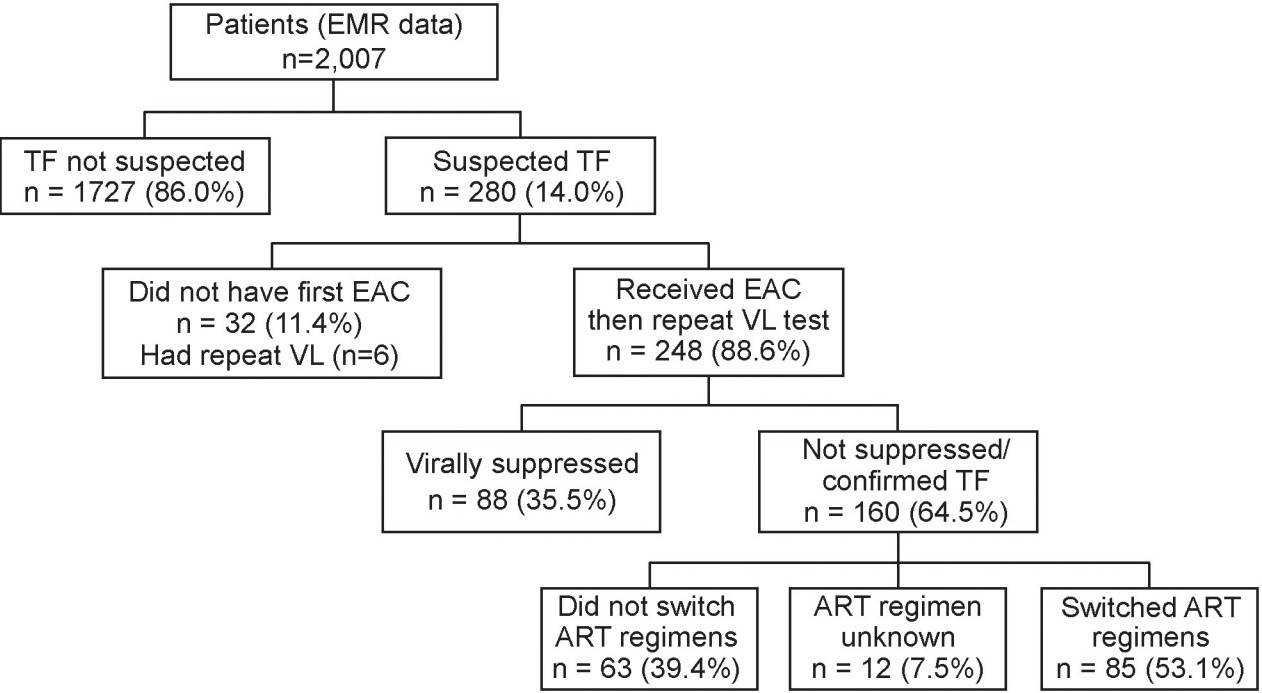

**Fig 1. Study flowchart.** This figure presents the number of participants enrolled and followed up in the study. Participants who did not have suspected treatment failure were not followed further. Participants with suspected treatment failure were followed up as they received EAC sessions. Those who received EAC sessions were followed up until they had a repeat VL. Participants who were suppressed following repeat VL were not followed up, while those who were not suppressed were either switched to a different ART regimen or not switched.

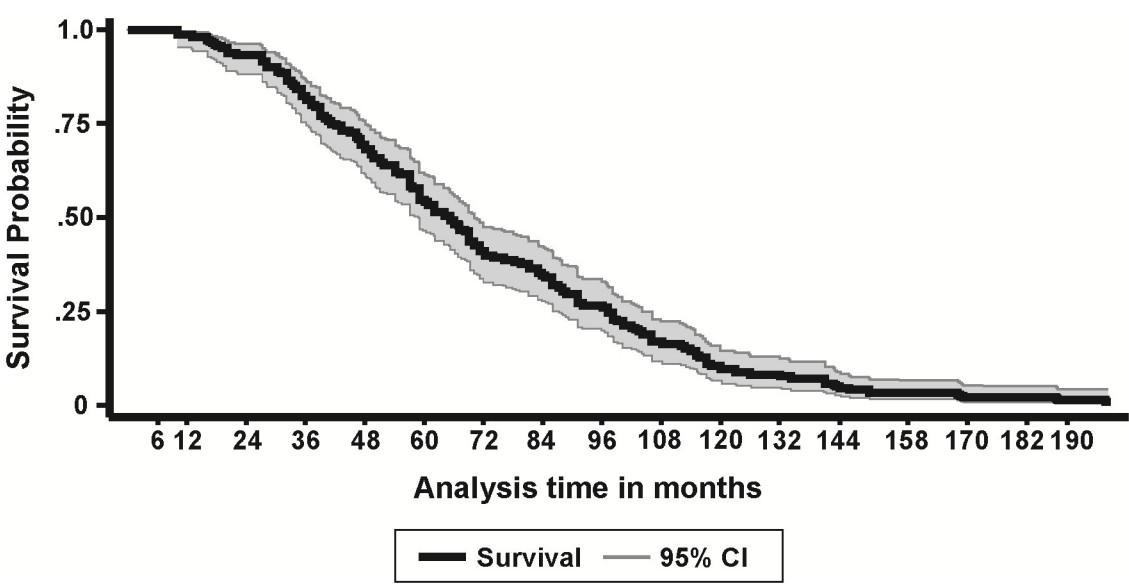

**Fig 2. Time to confirmed VTF.** Kaplan Meier cumulative probability of VTF following ART initiation at different time intervals, n = 160.

In the analysis of the client characteristics against VL suppression and non-suppression, VTF rates were higher among clients who were males, below 15 years of age at ART initiation, had attended dispensaries/health centers, experienced OIs, and experienced ART side effects (Table 2).

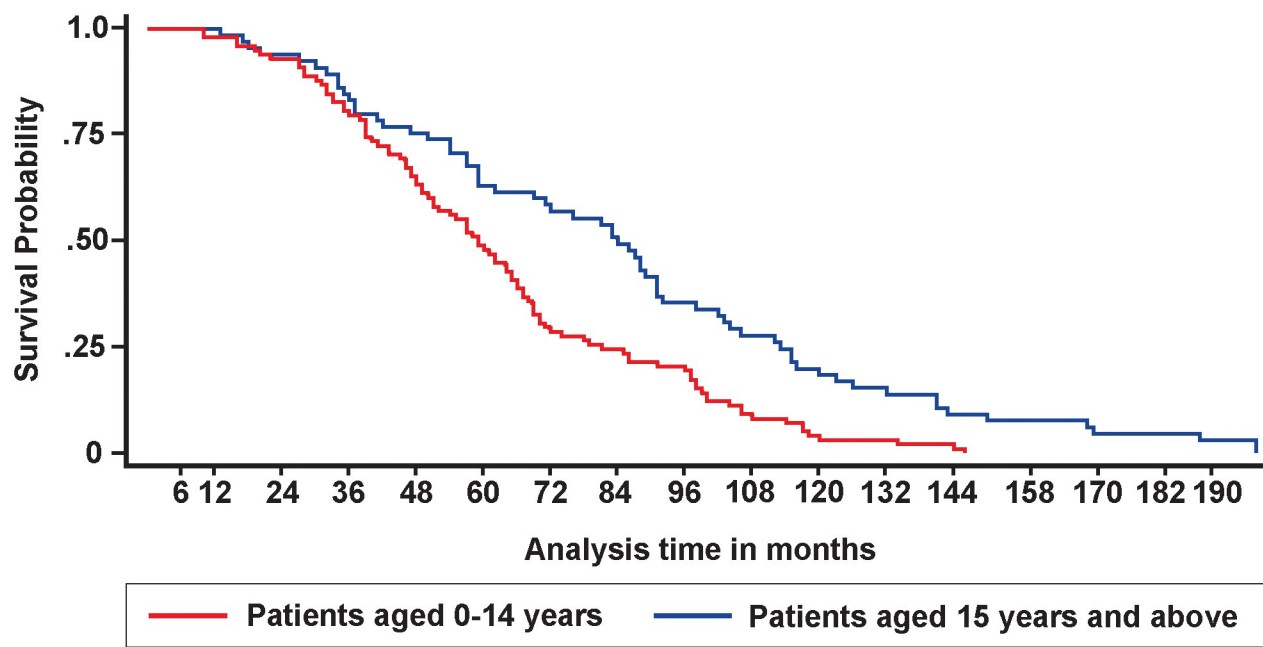

**Fig 3. Time to confirmed VTF from ART initiation by age.** Kaplan Meier cumulative probability of VTF following ART initiation at different time intervals by age group, patients aged 0–14 years (Red), patients aged 15 years and above (Blue), n = 160.

**Table 2. Characteristics of study participants by viral load; VL <1000 copies vs confirmed VL ≥1,000 copies/ml.**

| Characteristics | Level | VL <1000 copies/ml | VL ≥1000 copies/ml | P-value[a] |
|---|---|---|---|---|
| | | N = 1844 | N = 160 | |
| | | n (row %) | n (row %) | |
| Sex[b] | Male | 751 (90.6) | 78 (9.4) | 0.033 |
| | Female | 1084 (93.2) | 79 (6.8) | |
| Age at ART initiation, years[c] | 0–14 | 644 (84.8) | 115 (14.7) | <0.001 |
| | 15+ | 1180 (96.3) | 45 (3.7) | |
| Facility type | Level 2/3 | 324 (89.0) | 40 (11.0) | 0.019 |
| | Level 4/5 | 1520 (92.7) | 120 (7.3) | |
| Time to ART initiation, days | 0–30 | 791 (93.4) | 56 (6.6) | 0.052 |
| | >30 | 1053 (91.0) | 104 (9.0) | |
| ART regimen at initiation[d,e] | NRTI/NNRTI-based | 1645 (89.4) | 141 (88.7) | 0.118 |
| | PI-based | 139 (7.6) | 18 (11.3) | |
| | DTG-based | 56 (3.0) | 0 (0.0) | |
| ART regimen line[d] | 1st line | 1700 (92.3) | 141 (7.7) | 0.096 |
| | 2nd/3rd line | 140 (88.6) | 18 (11.4) | |
| Experienced at least one ART side effect | Yes | 357 (84.8) | 64 (15.2) | <0.001 |
| | No | 1487 (93.9) | 96 (6.1) | |
| WHO stage at ART initiation[d] | WHO I/II | 1304 (92.4) | 107 (7.6) | 0.283 |
| | WHO III/IV | 535 (91.0) | 53 (9.0) | |
| Experienced at least one OI | Yes | 364 (87.1) | 54 (12.9) | 0.001 |
| | No | 1480 (93.3) | 106 (6.7) | |
| CD4 at ART initiation, cells/ml of blood[f] | <200 | 282 (88.4) | 37 (11.6) | 0.062 |
| | ≥200 | 880 (91.9) | 78 (8.1) | |

This table presents the VTF rates by socio-demographic and clinical characteristics of the clients by viral load

[a] Chi-square test

[b] Data were missing for 12 clients

[c] Data were missing for 20 clients

[d] Data were missing for five clients

[e] Percentages were calculated by column, not row

[f] Data were missing for 727 clients

## Timing of services among patients failing treatment

Overall, the median time to ART initiation following HIV-positive diagnosis was 71 days (IQR: 9.5, 378). The median time to ART initiation was significantly higher among those aged 15 years and above, at 123 days (IQR: 37, 727), compared to children 0–14 years, at 65 days (IQR: 0, 251), p-value = 0.024. Time to elevated VL following ART initiation was lower among children 0–14 years at ART initiation at four years (IQR: 3, 7), compared to six years (IQR: 3, 8) for those who were 15 years and above at ART start (p-value: 0.051). The median time to first EAC after elevated VL was 60 days (IQR: 35, 118), and 50 days (IQR: 21, 153) among 0–14 year-olds and 15 years or older clients respectively, p = 0.319. The median time to complete at least three EAC sessions was 70 days(IQR: 48, 93), and 68 days (IQR: 53, 104) among 0–14 years and 15 years or older clients respectively, p = 0.575. Similarly, the median time to repeat VL from last EAC session was 143 days (IQR: 56, 303), and 159 days (IQR: 55, 296) among 0–14 years and 15 years or older clients respectively, p = 0.947.

**Table 3. Factors associated with confirmed VTF.**

| Factor | Levels | Crude HR [95% CI] | P-value | Adjusted HR [95% CI] | P-value |
|---|---|---|---|---|---|
| Sex[a] | Male | 1.363 [1.000, 1.857] | 0.050* | 1.174 [0.855, 1.613] | 0.322 |
| | Female | 1 | | 1 | |
| Age, years | 0 to 14 | 4.458 [3.167, 6.274] | <0.001*** | 4.420 [3.122, 6.320] | <0.001*** |
| | 15+ | 1 | | 1 | |
| Facility | Level 2/3 | 2.155 [1.513, 3.070] | <0.001*** | 1.874 [1.291, 2.721] | 0.001*** |
| | Level 4/5 | 1 | | 1 | |
| Time to ART initiation, days | 0–30 | 1 | | | |
| | >30 | 0.919 [0.668, 1.265] | 0.605 | na | |
| ART regimen at VTF[b] | NRTI/NNRTI-based | 1 | | | |
| | PI-based | 0.802 [0.563, 1.144] | 0.224 | na | |
| | DTG-based | 0.687 [0.222, 2.122] | 0.514 | na | |
| WHO at ART initiation | I/II | 1 | | 1 | |
| | III/IV | 0.788 [0.574, 1.084] | 0.143* | 0.823 [0.592, 1.143] | 0.245 |
| CD4 AT ART initiation, cells/ml [c] | <200 | 1.238 [0.840, 1.824] | 0.281 | na | |
| | > = 200 | 1 | | | |
| Experienced at least one side effect | Yes | 2.103 [1.541, 2.869] | <0.001*** | 2.434 [1.757, 3.371] | <0.001*** |
| | No | 1 | | 1 | |
| Experienced at least one opportunistic infection | Yes | 1.735 [1.253, 2.402] | 0.001*** | 1.807 [1.286, 2.537] | 0.001*** |
| | No | | | | |

This table presents the factors associated with confirmed VTF among clients. The Cox proportional hazard model was used to assess these factors

*** $p < .01$,

** $p < .05$,

* $p < .2$

VTF- - -Virologic Treatment Failure, HR- - -Hazard Ratio

[a] Data were missing for three clients

[b] Data was missing for one client

[c] Data were missing for 45 clients

### Factors associated with confirmed VTF

Univariable and multi-variable analysis were conducted to determine factors associated with VTF. In univariate analysis, factors associated with VTF were: being male, HR: 1.36 (95% CI: 1.0, 1.86); younger age at ART initiation, 0–14 years, HR: 4.46 (95% CI: 3.17, 6.27); receiving HIV services in level 2 and 3 facilities (health centers/dispensaries), HR: 2.16 (95% CI: 1.51, 3.07); having experienced side effects, HR: 2.10 (95% CI: 1.54, 2.87); and having experienced OIs, HR: 1.74 (95% CI: 1.25, 2.40).

In multi-variable analysis, the factors significantly associated with VTF were: age 0–14 years at ART initiation, adjusted hazard ratio (AHR): 4.42 (95% CI: 3.12, 6.32); experienced side effects, AHR: 2.43 (95% CI: 1.76, 3.37); and had experienced OIs, AHR: 1.81 (95% CI: 1.29, 2.54) (Table 3).

## Discussion

In summary, we found that confirmed VTF was identified among 8% of patients attending selected facilities in Homa Bay County, Kenya. VTF was associated with being less than 15 years at time of treatment initiation. VTF among children when starting ART was 14.7% (95% CI: 12.2%, 17.2%), significantly higher than 3.7% (95% CI: 2.6%, 4.7%) among adults.

Although the rate of VTF was higher among PLHIV who stated ART with CD4 <200 cells/mL (11.6% (95% CI: 8.1%, 15.1%)) compared to those who started ART with CD4 > = 200 cells/mL (8.1% (95% CI:6.4%, 9.9%)), this difference was not statistically significant, p = 0.062.

The failure rate of 3.7% among adults in this study was inconsistent with previous findings from studies conducted among adults in Ethiopia, Uganda, India, and Kenya [13, 14, 22–27], where rates of VTF among adults ranged from about 10% to as high as 34%. Many of these studies however, used the definition of two consecutive VL >1000 copies/ml$^3$ 3 months apart, without the condition of adherence counseling, somewhat different to the definition used in our study. The prevalence of virologic failure amongst children was comparable to what has been reported in prior studies in low and middle-income counties, which found rates ranging from 10% to as high as 37% [9, 28, 29]. A 2018 study by Kadima et al. was conducted in the same region as this study, Homa Bay County, Kenya, and found a 37% failure rate among children. The variation between VTF rates among children in the prior and current study may be related to the populations studied, the difference in age cut off, when the studies were conducted, and what definition of VTF was used [30]. Additionally, we had more optimal regimens, like DTG, introduced within the study period. We may have obtained a more conservative estimate of a failure rate, as our definition of confirmed VTF was based on VL testing after EAC, where patients were still failing treatment. We also defined VTF as VL ≥1000 copies/ml, while some studies may have defined VTF as VL >400 copies/ml.

The majority of those failing treatment were children or adolescents when starting ART, groups known to have high VTF rates [31]. Relatively higher VLs in early childhood may be associated with VTF in children [32]. VTF may be related also to drug resistance (particularly NNRTIs), as well as challenges with administration of the drug and possible unpalatability (e.g. lopinavir/ritonavir) [33]. In accordance with WHO guidelines, countries, including Kenya, are switching to pediatric DTG, which is highly potent, has a high barrier to resistance, is easier to administer, and is more palatable [34]. High VL as a potential indicator of VTF may be useful as action may be taken earlier to reduce the possibility of VTF, and consequent negative consequences. Among adolescents, poor adherence may be more of a factor in VTF. A recent systematic review and metanalysis on VTF found poor adherence to be a factor in ART VTF [11]. This speaks to the need for strengthening innovative interventions, such as "Operation Triple Zero", which empowers adolescents and young people to take control of their health [35]. Additional innovative strategies to address sub-optimal adherence to ART among adolescents would be establishment of youth-targeted services, exclusively focused for youth [36]. The use of video directly-observed therapy in individuals who present with a first elevated VL to improve adherence and viral suppression should be explored also [37].

Although our study had more females, the rate of VTF was higher among males. This finding is consistent with other studies [38–40], possibly because males have poorer health seeking behavior than females [41]. In this study, while 81% of females with VTF completed three EAC sessions, only 73%% of males completed their sessions. Strategies such as male friendly clinics, extended clinic hours, and the provision of peer navigation could be strengthened to improve men's attendance to EAC sessions [36, 42].

Those who experienced OIs were one and half times more likely to experience VTF (AHR: 1.81 [95% CI: 1.29, 2.54]); this likely reflects the more compromised immunologic status of individuals who experienced VTF. This factor has been reported in a systematic review as being associated with VTF [11]. Low CD4 count and experience of OIs are correlated with significantly impaired immune systems and late presentation for treatment, where even with appropriate treatment, there is an enhanced risk of VTF [43, 44]. Almost all participants with VTF in our study were screened for TB and received IPT (99.4%), which is higher compared to findings of a study in Kenya that reported 85% IPT uptake among PLHIV [45].

Likewise, patients who experienced ART side effects were more than twice as likely to develop VTF (AHR: 2.43 [95% CI: 1.76, 3.37]) compared with those who did not experience any side effects. ART side effects have been shown to be associated with increased risk of non-adherence and treatment discontinuation leading to VTF [46, 47].

Patients attending level 2/3 facilities, (dispensaries and health centers), were nearly 1.9 times (95% CI: 1.29–2.72) more likely to have VTF. These facilities are usually the first entry into the health care system, patients attending may already be very ill with impaired immune systems as indicated by the occurrence of OIs. Additionally, staff at these facilities may not be as well trained as those at higher level facilities, so many not manage the patients or refer to higher level facilities in a timely manner.

Late initiation of ART (>30 days) did not appear to be associated with a greater likelihood of VTF (HR: 0.91[95% CI: 0.67–1.27]), contrary to studies affirming the importance of early treatment [48, 49]. It may be that an impaired immune system as indicated by the occurrence of OIs, better reflects the risk of developing VTF, rather than late ART initiation.

Over half of the patients with VTF were failing treatment between 24 and 60 months after ART initiation. This is comparable to what was reported by Wools-Kaloustian from the IeDEA pediatric cohort of 57%, when the failure and attrition rates of 31.6% and 25.9%, respectively, are combined [50]. In our study, VTF at 12 months was 5.6%, as compared with 7.7% in the IeDEA cohort. Our study included both children and adults, and, while children at time of ART initiation only made up 39% of the overall sample, they accounted for 71.9% of those who were diagnosed with VTF.

At the time of data collection, 63/160 (39.4%) of the clients with confirmed VTF had not been switched to a different regimen. This emphasizes the need for treatment programs to adhere to treatment guidelines and promptly switch patients with confirmed VTF to optimal regimens to avoid the accumulation of resistant mutations [51].

A limitation of our study is that it was retrospective and data were abstracted from routine medical records. For a number of variables, data were either missing, incomplete, inconclusive, or inaccurate. For example, currently not all patients get a CD4 count done at ART initiation. For our study participants, CD4 count data at ART initiation were missing for 36.2% of participants with VL data. This may have biased the association between CD4 count and VTF, by over or underestimating it. A second limitation is that our study included retrospective data abstraction on individuals on ART at the time of data abstraction, and hence did not include deaths, loss to follow ups, or transfers, which could have biased our estimates. We based our diagnosis of VTF on a repeat VL following EAC, yet some of the participants did not get a repeat VL. However, strengths of the study include the relatively large sample size and that it was undertaken in an area of high HIV prevalence in Kenya. Also, our study was conducted under programmatic conditions (real-world conditions) using routine medical records at public health facilities, demonstrating the burden of VTF in real-world conditions, therefore, maximizing generalizability.

## Conclusion

Children, attendees of level 2/3 health facilities, patients with a history of OIs, and those experiencing treatment side-effects, are at risk of VTF. There is need for increased focus on children and adolescents on screening for drug resistance, administration of and adherence to medication, and on effective information and education on side-effects for patients overall. Additionally, there is need for increased training and support for health personnel at primary level care facilities in screening for potential treatment failure, appropriate and timely referral, and adherence guidance.

## Supporting information

**S1 Data. Treatment failure data.** This data file contains the data that were analyzed and presented in this paper.
(XLS)

## Acknowledgments

We greatly appreciate the assistance provided by the study investigators, the EGPAF study coordination and data team, and other EGPAF staff whose work and dedication made this study possible. The authors would like to thank the team of research assistants for their essential role in this study and the study participants, without whom, this research would not be possible. We also wish to acknowledge the essential cooperation and collaboration of the Homa Bay County Ministry of Health and CDC for their technical support throughout this study. We thank Adrienne Hayes and Shannon Viana from EGPAF, DC, for the final review and editing of this manuscript.

## Author Contributions

**Conceptualization:** Rose Masaba, Godfrey Woelk, Boniface Ochanda, Gordon Okomo.

**Data curation:** Rose Masaba, Stephen Siamba.

**Formal analysis:** Stephen Siamba.

**Investigation:** Rose Masaba.

**Methodology:** Rose Masaba.

**Project administration:** Rose Masaba, James Ndimbii, Millicent Ouma, Jacob Khaoya, Abraham Kipchirchir.

**Supervision:** James Ndimbii, Millicent Ouma.

**Validation:** Stephen Siamba.

**Visualization:** Stephen Siamba.

**Writing – original draft:** Rose Masaba.

**Writing – review & editing:** Stephen Siamba, James Ndimbii, Millicent Ouma, Jacob Khaoya, Abraham Kipchirchir, Boniface Ochanda, Gordon Okomo.

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
