## [Decision Letter · Decision Letter 0]

25 Oct 2022

PGPH-D-22-01209

Antiretroviral treatment failure and associated factors among people living with HIV on therapy in Homa Bay, Kenya: A retrospective study

Dear Dr. Masaba,

Thank you for submitting your manuscript to PLOS Global Public Health. After careful consideration, we feel that it has merit but does not fully meet PLOS Global Public Health’s publication criteria as it currently stands. Therefore, we invite you to submit a revised version of the manuscript that addresses the points raised during the review process.

We look forward to receiving your revised manuscript.

Kind regards,

Sanghyuk S Shin

Academic Editor

Journal Requirements:

a. Please clarify all sources of funding (financial or material support) for your study. List the grants (with grant number) or organizations (with url) that supported your study, including funding received from your institution. 

b. State the initials, alongside each funding source, of each author to receive each grant.

c. State what role the funders took in the study. If the funders had no role in your study, please state: “The funders had no role in study design, data collection and analysis, decision to publish, or preparation of the manuscript.”

d. If any authors received a salary from any of your funders, please state which authors and which funders.

2. We have noticed that you have uploaded Supporting Information files, but you have not included a list of legends. Please add a full list of legends for your Supporting Information files after the references list. 

3. In the online submission form, you indicated that "The datasets generated and/or analyzed during the current study are presented in the figures and tables in the manuscript. The analytic datasets are available from the corresponding author on request.". All PLOS journals now require all data underlying the findings described in their manuscript to be freely available to other researchers, either 1. In a public repository, 2. Within the manuscript itself, or 3. Uploaded as supplementary information.

Additional Editor Comments (if provided):

While the manuscript is clearly written, there are a few critical flaws in statistical analysis that should be addressed.

- The analysis does not take into account loss-to-follow-up (i.e. right censoring). Participant data were examined during a 4 month period (Sep 2020 – January 2021). Participants who started ART closer to the end of data collection would, therefore, have lower opportunity for VTF. This could lead to significant bias in classification of suspected TF and, thereby, bias the estimated associations of predictors. Please consider redoing the analysis to account for this bias, such as with time-to-event outcome methods.

- The definition of VTF is actually those with confirmed elevated VL after EAC. The authors should justify why this outcome is meaningful. It seems that the conservative outcome classification would be to count everyone with suspected TF as VTF unless they have confirmed viral suppression at the post-EAC measure. If the goal is to identify predictors of VTF among those who receive EAC, then the denominator for that analysis should be 280 people who received EAC.

Minor comments

- In the Introduction or the Methods, the authors should describe what EAC entails.

- Line 104. Please justify why the study included only those who were on treatment for over 6 months.

- Line 104. Were all patients who were in the database with treatment over 6 months included in the sample? Please state if other inclusion or exclusion criteria were used to create your sample.

- Line 109. While the timing of the data collection is reported, the range of dates for the retrospective data included in the sample is unclear. Please report the distribution of data with respect to treatment start dates.

- Line 124. Please justify why VTF was defined based on VL measurements after 3 months of EAC.

- Please report the number of people who had no VL measurements after 3 months of EAC classified. How were these handled in the analysis?

- Lines 160-162. I found the % advanced disease reporting confusing. It is reported as 30%in line 160 and then 24.9% in line 163. Is this because the first number included WHO stage III/IV? Or because the second number excluded missing CD4? I suggest deciding on one way to present the % of advanced immune suppression and simply presenting that % only.

- Line 174. The VTF rates are different than Table 3. Please make them consistent.

- What was the distribution of time to VTF from EAC?

- Table 2 contains the same data as Table 3 Column 4, except in column percent vs. row percent. Suggest deleting Table 2.

- Line 221. The term “predictor” is used, but some of the variables include those that are likely to be the result of VTF, not antecedent predictors (e.g. CD4 <200 cells/ml). Please justify the inclusion of each variable as “predictor” in the final multivariable model from a conceptual perspective, while avoiding over-adjustment bias. (See Schisterman EF, Cole SR, Platt RW. Overadjustment Bias and Unnecessary Adjustment in Epidemiologic Studies: Epidemiology. 2009 Jul;20(4):488–95.)

- Discussion. When providing context with prior studies, please explain how those studies defined VTF and whether they are comparable to the definition used in this manuscript.

- Line 254. Please explain the usefulness of noting the association between higher VLs and VTF, since VTF is defined by high VLs.

- Lines 267-268. I don’t believe EAC completion was presented in Results. Please include these data in the Results and use the Discussion to explain the context and implications of those results.

- Lines 274-275. Please note that low CD4 counts could be the result of the VTF, not the cause of it. Please explain whether the study design allowed for the conclusions that low CD4 may have a causal association (i.e. “predictor”) with VTF.

Reviewers' comments:

Reviewer's Responses to Questions

**Comments to the Author**

1. Does this manuscript meet PLOS Global Public Health’s publication criteria? Is the manuscript technically sound, and do the data support the conclusions? The manuscript must describe methodologically and ethically rigorous research with conclusions that are appropriately drawn based on the data presented.

Reviewer #1: Yes

2. Has the statistical analysis been performed appropriately and rigorously?

Reviewer #1: No

3. Have the authors made all data underlying the findings in their manuscript fully available (please refer to the Data Availability Statement at the start of the manuscript PDF file)?

Reviewer #1: No

4. Is the manuscript presented in an intelligible fashion and written in standard English?

Reviewer #1: Yes

5. Review Comments to the Author

Reviewer #1: Title “Antiretroviral treatment failure and associated factors among people living with HIV on therapy in Home Bay, Kenya : A retrospective study”

General comments

I would like to congratulate the authors for the much efforts made to produce this manuscript on “Antiretroviral treatment failure and associated factors among people living with HIV on therapy in Home Bay, Kenya: A retrospective study”. The authors made an attempt to answer the research question at satisfactory level, use the appropriate study design, presents the results and discuss the findings in light of the current literature. The authors have done a very good study. The aim of the study is clear. However, the manuscript requires minor revisions, which the authors need to address before publication.

1. Abstract

Line 24: Kindly state which region are referring to.

Line 36-37: Be consistent in your reporting. Kindly refer to line 35 (reporting proportion with corresponding 95% CIs). Consider removing these 95% CI relative to proportions given they do not read well.

Line 38-40: It is not clear when these CD4 counts were measured. I guess these are CD4 counts at ART-initiation. If so then explicitly state (Patients who initiated ART with CD4 counts < 200 cells/mL were 2.41 times). In addition, add their corresponding 95% CIs as the reader needs to assess the level of significance and the direction of the association.

Line 44: I fail to understand this part of your conclusion. In fact, HIV program implementers and Health care workers should focus on PLHIV with these characteristics in order to prevent and for early identification of ART treatment failure.

2. Background

Line 48-50: Kindly verify these statistics. These are estimates are for 2020 and the reporting time is mixed up. Updates these statistics with 2021. Kindly see the link below:

https://www.unaids.org/en/resources/fact-sheet

Line 52: These figures are obsolete and outdated, kindly update these using the link provided in the comment above.

Line 83-85: While the second part of the study aim is clear, and it has been adequately addressed throughout this paper. However, I have substantial concern regarding the following: “To better describe the HIV treatment cascade”. This was not addressed in this study.

3. Methods

Line 88-89: Provide further details on the study design, especially on the direction of inquiry.

Line 106-107: Is this a desired sample size based on sample size calculation or a sample size chosen arbitrarily (conveniently) based on resource and time. Kindly clarify

Line 110-111: It is important to provide further details on how the random systematic sampling was conducted leading to a selection of patients included in the analysis. A single sentence will suffice

Line 129-131: Your primary outcome is time-to-event data, reason why Kaplan-Meier method was used to estimate the probabilities of VTF. However, using logistic regression to identify factors associated with VTF is inappropriate for time-to-event data, given logistic regression provides biased estimates when dealing with time-to-event data. To identify factors associated with VTF, Cox-regression methods also known as proportional hazards regression is the suitable/adequate method for the type of data. Kindly re-run these models using Cox-regression methods.

4. Results

Line 147-148: I am not knowledgeable of the of Kenyan context, do you mean individuals aged 0-14 were married?

Line 161: Which method did you use to assess advanced HIV disease ( CD4 count or WHO staging system) ? Looking at table 1, 30% of patients had AHD and this is based on CD4 count. If this is the case, then delete WHO stage III and IV in the sentence.

Line 178: This is not good practice for reporting, DO NOT report mean with 95% confidence intervals , instead report:

1. Mean with standard deviation (SD).

2. Median with IQR (as you have been reporting)

On the lighter note, generally, time-to-event data are skewed or asymmetrical (not all the time), kindly check the distribution of your data before reporting the mean time to VTF, otherwise use median time to VTF.

Line 220: Refer to my comment in line 129-131. The whole section will change in terms of interpreting findings. I am not expecting substantive changes in the overall results or key findings. There is a sizable number of papers to why Cox proportional hazards models are more suitable instead logistic regression models in when dealing with time-to-event data.

https://www.nature.com/articles/ejhg200859

3. Discussion

Line 237-239: This is critical part of the discussion section as it summarises key findings of the paper. Kindly add VTF was significantly higher among PLHIV who stared ART with CD4 <200 cells/mL.

Line 247: Change population observed to the "populations studied". Furthermore, the variation observed between your study and Kadima et al. could be partly explained by differences in age cut-off for children.

Line 258-259: Additional innovative strategies to address sub-optimal adherence to ART among adolescents would be establishment of youth-targeted services, exclusively to youth.

https://onlinelibrary.wiley.com/doi/full/10.1002/jia2.25854

Line 269: I suggest adding a more recent evidence from South Africa.

https://onlinelibrary.wiley.com/doi/full/10.1002/jia2.25854

Line 298: What is the possible impact of missing CD4 count data on main findings?

Missing data often leads to under or over-estimation (could bias your estimates) of the measures of effects (in the context of your study, this could lead an over or under-estimation of the association between CD4 count and VTF)

Line 303-304: Consider to add another strength: "Our study was conducted under programmatic conditions (real-world conditions) using routine medical records at public health facilities, demonstrating the burden of VTF in real-world conditions, therefore, maximizing generalizability".

6. PLOS authors have the option to publish the peer review history of their article (what does this mean?). If published, this will include your full peer review and any attached files.

**Do you want your identity to be public for this peer review?** For information about this choice, including consent withdrawal, please see our Privacy Policy.

Reviewer #1: **Yes: **Marcel Kanyinda Kitenge

---

## [Decision Letter · Decision Letter 1]

17 Jan 2023

PGPH-D-22-01209R1

Antiretroviral treatment failure and associated factors among people living with HIV on therapy in Homa Bay, Kenya: A retrospective study

Dear Dr. Masaba,

Thank you for submitting your manuscript to PLOS Global Public Health. After careful consideration, we feel that it has merit but does not fully meet PLOS Global Public Health’s publication criteria as it currently stands. Therefore, we invite you to submit a revised version of the manuscript that addresses the points raised during the review process.

The reviewer has uploaded a PDF of additional comments that should be addressed. Please review these comments carefully and respond to them in your next submission.

We look forward to receiving your revised manuscript.

Kind regards,

Sanghyuk S Shin

Academic Editor

Journal Requirements:

2. We have noticed that you have uploaded Supporting Information files, but you have not included a list of legends. Please add a full list of legends for your Supporting Information files after the references list. 

Additional Editor Comments (if provided):

Reviewers' comments:

Reviewer's Responses to Questions

**Comments to the Author**

1. If the authors have adequately addressed your comments raised in a previous round of review and you feel that this manuscript is now acceptable for publication, you may indicate that here to bypass the “Comments to the Author” section, enter your conflict of interest statement in the “Confidential to Editor” section, and submit your "Accept" recommendation.

Reviewer #1: All comments have been addressed

2. Does this manuscript meet PLOS Global Public Health’s publication criteria? Is the manuscript technically sound, and do the data support the conclusions? The manuscript must describe methodologically and ethically rigorous research with conclusions that are appropriately drawn based on the data presented.

Reviewer #1: Yes

3. Has the statistical analysis been performed appropriately and rigorously?

Reviewer #1: Yes

4. Have the authors made all data underlying the findings in their manuscript fully available (please refer to the Data Availability Statement at the start of the manuscript PDF file)?

Reviewer #1: Yes

5. Is the manuscript presented in an intelligible fashion and written in standard English?

Reviewer #1: Yes

6. Review Comments to the Author

Reviewer #1: (No Response)

7. PLOS authors have the option to publish the peer review history of their article (what does this mean?). If published, this will include your full peer review and any attached files.

**Do you want your identity to be public for this peer review?** For information about this choice, including consent withdrawal, please see our Privacy Policy.

Reviewer #1: **Yes: **Marcel Kitenge

Please see the reviewer's comments submitted in PDF format.

---

## [Editor Report · Decision Letter 2]

2 Feb 2023

Antiretroviral treatment failure and associated factors among people living with HIV on therapy in Homa Bay, Kenya: A retrospective study

PGPH-D-22-01209R2

Dear Dr Masaba,

We are pleased to inform you that your manuscript 'Antiretroviral treatment failure and associated factors among people living with HIV on therapy in Homa Bay, Kenya: A retrospective study' has been provisionally accepted for publication in PLOS Global Public Health.

Best regards,

Sanghyuk S Shin

Academic Editor
